# REBOOT: Reuse Data for Bootstrapping Efficient Real-World Dexterous Manipulation

**Zheyuan Hu**[1*]**, Aaron Rovinsky**[1*]**, Jianlan Luo**[1]**, Vikash Kumar**[2]**, Abhishek Gupta**[3]**, Sergey Levine**[1]

[1] UC Berkeley  [2] Meta AI Research  [3] University of Washington

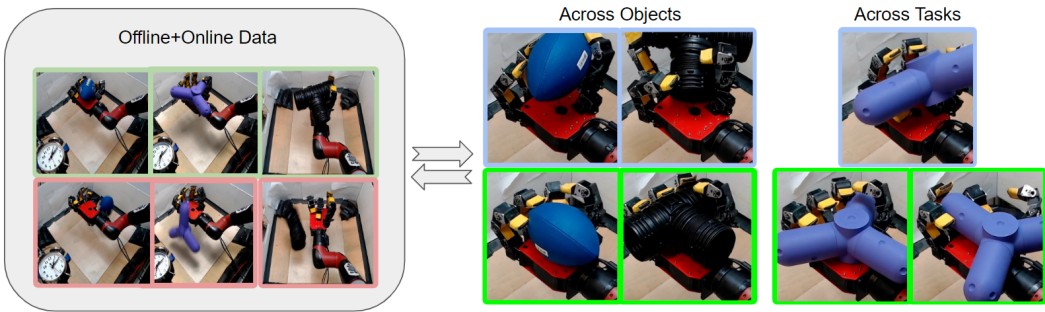

Figure 1: REBOOT achieves **2X** sample efficiency boost on learning a variety of contact-rich real-world dexterous manipulation skills on three different objects autonomously by bootstrapping on prior data across different objects and tasks with sample-efficient RL and imitation learning-based reset policies.

**Abstract:** Dexterous manipulation tasks involving contact-rich interactions pose a significant challenge for both model-based control systems and imitation learning algorithms. The complexity arises from the need for multi-fingered robotic hands to dynamically establish and break contacts, balance non-prehensile forces, and control large degrees of freedom. Reinforcement learning (RL) offers a promising approach due to its general applicability and capacity to autonomously acquire optimal manipulation strategies. However, its real-world application is often hindered by the necessity to generate a large number of samples, reset the environment, and obtain reward signals. In this work, we introduce an efficient system for learning dexterous manipulation skills with RL to alleviate these challenges. The main idea of our approach is the integration of recent advances in sample-efficient RL and replay buffer bootstrapping. This combination allows us to utilize data from different tasks or objects as a starting point for training new tasks, significantly improving learning efficiency. Additionally, our system completes the real-world training cycle by incorporating learned resets via an imitation-based pickup policy as well as learned reward functions, eliminating the need for manual resets and reward engineering. We demonstrate the benefits of reusing past data as replay buffer initialization for new tasks, for instance, the fast acquisition of intricate manipulation skills in the real world on a four-fingered robotic hand. (Videos: https://sites.google.com/view/reboot-dexterous)

**Keywords:** Dexterous Manipulation, Reinforcement Learning, Sample-Efficient

## 1 Introduction

Dexterous manipulation tasks involving contact-rich interaction, specifically those involving multi-fingered robotic hands and underactuated objects, pose a significant challenge for both model-based control systems and imitation learning algorithms. The complexity arises from the need for multi-

---

[*]Both authors contributed equally

7th Conference on Robot Learning (CoRL 2023), Atlanta, USA.

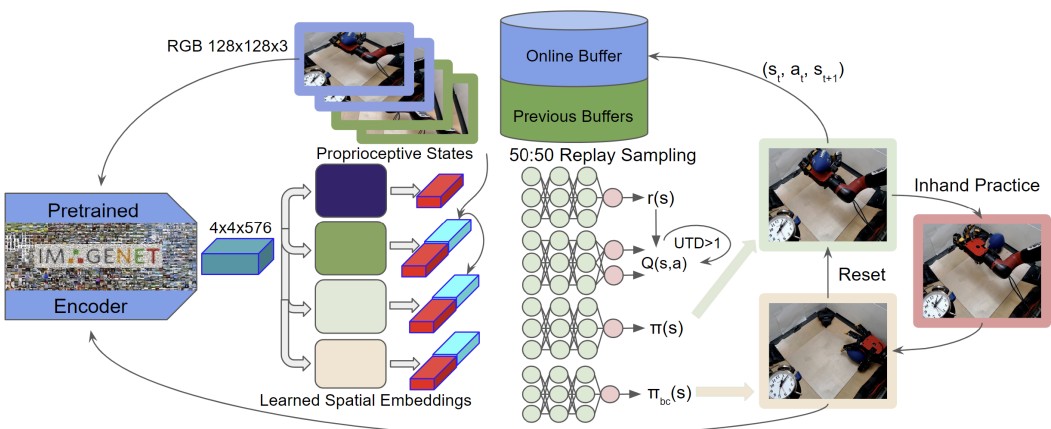

Figure 2: **REBOOT** System Overview: Our method learns various dexterous manipulation skills in the real world using raw image observations. This is enabled by using sample-efficient RL and bootstrapping with data from other tasks and even other objects, with autonomous resets.

fingered robotic hands to dynamically establish and break contacts, balance non-prehensile forces, and control a high number of degrees of freedom. Reinforcement learning (RL) offers a promising solution for such settings. In principle, RL enables a robot to refine its manipulation skills through a process of trial-and-error, alleviating the requirement for strong modeling assumptions. However, making RL methods practical for learning such complex behaviors directly in the real world presents a number of obstacles. The main obstacle is sample efficiency: particularly for tasks that require complex interactions with many possibilities for failure (e.g., in-hand reorientation where the robot might drop the object), the number of trials needed for learning a skill with RL from scratch might be very high, requiring hours or even days of training. Additionally, real-world learning outside of the lab requires the robot to perform the entire training process using its own sensors and actuators, evaluating object state and rewards using camera observations, and resetting autonomously between trials. Because of these challenges, many prior works on RL for dexterous manipulation have explored alternative solutions, such as sim-to-real transfer [1, 2, 3], imitation learning [4, 5, 6], or the use of tools like motion capture [7, 2] or separately-engineered reset mechanisms [8, 9].

In this paper, we instead propose a system that is designed to make direct RL in the real world practical without these alternatives, so as to take a step toward robots that could one day learn under assumptions that are sufficient for autonomously acquiring new skills in open-world settings, even outside the laboratory. This means that the entire learning process must be conducted using the robot's own sensors and actuators, without simulation or additional instrumentation, and be efficient enough to learn skills quickly. We posit that a key enabling factor for this goal is to reuse data from past skills, and we instantiate this with a simple buffer initialization method, where the replay buffer of each skill is initialized with data from other tasks or even other objects. In combination with a vision-based method for learning reward functions from user-provided images and a learned reset procedure to automatically pick up an object between trials, we demonstrate that our system enables a robotic hand to learn in-hand reorientation skills in just a few hours of fully autonomous training, using only camera observations and joint encoder readings.

Our main contribution is **REBOOT**, a system to **Re**use Data for **Boot**strapping Real-World Dexterous Manipulation, which we illustrate in Figure 2. By simply preloading the replay buffer using prior data from other objects and tasks, our system avoids starting from scratch for every new task. By combining recent advances in sample-efficient online RL [10] with buffer initialization to bootstrap learning from prior tasks and objects, we show that in-hand manipulation behaviors can be learned in a few hours of autonomous practicing.

We additionally use learned reset skills to make training autonomous, and extend adversarially learned rewards to handle our buffer initialization method, allowing users to specify tasks with a few examples of desired object poses and without manual reward engineering. Some of the skills

learned by our system, shown in Figure 3, include in-hand reorientation of a three-pronged object, handling a T-shaped pipe, and manipulating a toy football.

## 2   Related Work

A number of designs for anthropomorphic hands have been proposed in prior work [11, 12, 13]. Prior learning-based methods to control such hands utilize trajectory optimization [14, 15], policy search [16, 17, 18], demonstration-based learning [19, 20, 21, 22], simulation to real-world transfer [3, 23, 24, 25], reinforcement learning directly in the real world [26, 8, 27, 28, 29], or a combination of these approaches [30].

Most of the aforementioned works leveraged accurate simulations or engineered real-world state-estimation systems to provide compact state representations. In contrast, we seek to learn visuomotor policies autonomously and entirely in the real world without access to privileged state information, under assumptions that more closely reflect what robots might encounter when learning "on the job" outside of laboratory settings. Prior work has explored learning these policies in simulation [31, 32], where autonomy is not of concern due to the ability to reset the simulated environment. Most real-world methods either rely on instrumentation for state estimation [28] or deal with simpler robots and tasks [27]. An important consideration in our system is the ability to specify a task without manual reward engineering. Although task specification has been studied extensively, most prior works make a variety of assumptions, ranges from having humans-provided demonstrations for enabling imitation learning [4, 33, 34], using inverse RL [35, 36, 37], active settings where users can provide corrections [38, 39, 40], or ranking-based preferences [41, 42]. Our in-hand RL training phase learns from raw high-dimensional pixel observations in an end-to-end fashion using DrQ[43] and VICE[44], although our system could use any reward inference algorithm based upon success examples [45]. With users defining the manipulation task by providing a small number of image goals instead of full demonstrations, our method not only removes the barrier to orchestrate high-dimensional finger motions [46, 47] but also accelerates robot training progress by offering sufficient reward shaping for RL in real-world scenarios without per-task reward engineering. While AVAIL [29] also learns dexterous manipulation skills from raw images, we show in our comparison that our system is faster, and our buffer initialization approach significantly speeds up the acquisition of in-hand manipulation skills compared to starting from scratch.

Buffer initialization has also been employed by Smith et al. [48] in the context of transfer learning for robotic locomotion, where a similar approach was used to create a curriculum for locomotion skills or adapt to walking on new terrains. Our method differs in several significant ways. First, our method learns from raw image observations with learned reward functions defined through a few example images, whereas [48] uses hand-programmed rewards. Second, our focus is on learning intricate dexterous manipulation skills from scratch in the real world, whereas [48] uses initialization in simulation. Although the methodology is closely related, our proposed system extends the methodology in significant ways, enabling the use of vision and learned rewards in a very different domain.

Reset-free learning is essential for autonomous real-world training of dexterous skills (see [49] for a review of reset-free methods). Most of the prior works [27, 28, 29, 50, 51, 52, 53, 54] rely on "backward" policies to reset the environment so the "forward" policy can continue learning the task of interest. Similarly, we divide training into two phases due to different skills having unique demands for control complexities and user-provided supervision. Specifically, the skill needed to pick up objects in reset is better studied and developed for immediate usage through imitating user-provided demonstrations [55].

## 3   Robot Platform and Problem Overview

In this work, we use a custom-built, 4-finger, 16-DoF robot hand mounted to a 7-DoF Sawyer robotic arm for dexterous object manipulation tasks. Our platform is shown in Figure 3. Our focus is on

learning in-hand reorientation skills with reinforcement learning. During the in-hand manipulation phase, the RL policy controls the 16 degrees of freedom of the hand, setting target positions at 10 Hz, with observations provided by the joint encoders in the finger motors and two RGB cameras, one overhead and another embedded in the palm of the hand. To facilitate autonomous training, we also use imitation learning to learn a reset policy to pick up the object from the table in-between in-hand manipulation trials. This imitation policy uses a 19-dimensional action space, controlling the end effector position of the wrist and 16 finger joints to pick up the object from any location.

Our tasks are parameterized by images of desired object poses in the palm of the hand. Since the reset policy can grasp the object in a variety of poses, the in-hand policy must learn to rotate and translate the object carefully to achieve the goal pose. We train and evaluate our method entirely in the real world. In the following sections, we describe how data from different objects can be used to bootstrap new manipulation skills for more efficient learning.

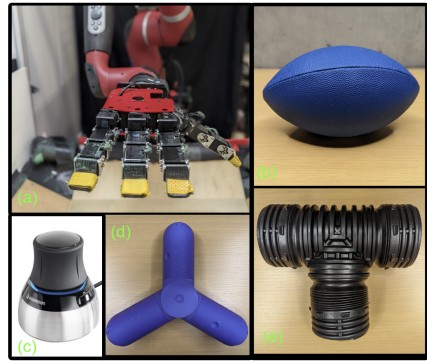

Figure 3: Depiction of our hardware platform and tasks. **(a)** custom-built 16 DoF robotic hand **(c)** teleoperation using the 3-D mouse, to interact with the following objects in-hand **(b)** blue football, **(d)** 3-pronged valve, **(e)** T-shaped pipe.

## 4 Reinforcement Learning with Buffer Initialization

In this work, we propose a system for efficiently learning visuomotor policies for dexterous manipulation tasks via bootstrapping with prior data. We describe our learning method and real-world considerations for our system in the following subsections.

**Problem setting.** Our method leverages the framework of Markov decision processes for reinforcement learning as described in [56]. In RL, the aim is to learn a policy $\pi(a_t|s_t)$ that obtains the maximum expected discounted sum of rewards $R(s_t, a_t)$ under an initial state distribution $\rho_0$, dynamics $\mathcal{P}(s_{t+1}|s_t, a_t)$, and discount factor $\gamma$. The formal objective is as follows:

$$J(\pi) = \mathbb{E}_{\substack{s_0 \sim \rho_0 \\ a_t \sim \pi(a_t|s_t) \\ s_{t+1} \sim \mathcal{P}(s_{t+1}|s_t, a_t)}} \left[ \sum_{t=0}^{T} \gamma^t R(s_t, a_t) \right] \tag{1}$$

The particular reinforcement learning algorithm that we build on in this work is RLPD [10], a sample-efficient RL method that combines a design based on soft actor-critic (SAC) [57] with a number of design decisions to enable fast training. This approach trains a value function $Q^\pi(s_t, a_t)$ in combination with an actor or policy $\pi(a_t|s_t)$, though in principle our system could be compatible with a variety of off-policy RL algorithms that utilize a replay buffer. For more details on RLPD, we refer readers to prior work [10].

**Reinforcement learning with buffer initialization.** While using a sample-efficient RL algorithm such as RLPD to acquire in-hand manipulation skills can be feasible in the real world, the training process can take a very long time (see Section 5). A central component of our system design is to utilize data from other tasks or even other objects to bootstrap the acquisition of new in-hand manipulation skills. In our experiments, we will show that a very simple procedure can make this possible: for every RL update, we sample half the batch from the growing buffer for the current task, and half the batch from a buffer containing the experience from all of the prior tasks. Thus, if $n-1$ skills have been learned, to learn a new $n$-th skill, we pre-load the replay buffer with trajectories from each of the $n-1$ prior skills and sample half of each training batch from prior data and the other half from the new agent's online experience. This 50-50 sampling method has been used in some prior works, including

RLPD [10, 58], in order to initialize online RL with offline data from *the same task*. However, in our system, we adapt this procedure to bootstrap a behavior from *other* skills. Since all of the tasks use visual observations, the generalization ability of the value function and policy networks can then learn to make use of this prior experience to assist in learning the new task. Note that it is not at all obvious that prior experience like this should be directly useful, as other tasks involve visiting very different states or manipulating different objects. However, if the networks are able to extract for example a general understanding of contacts or physical interactions, then we would expect this to accelerate the acquisition of the new task.

**Demonstration-based reset-free learning.** In-between in-hand manipulation trials, the robot may drop the object and need to pick it back up again to attempt the task again. To automate training, we must also acquire an autonomous pick-up policy to serve as a reset mechanism for the in-hand task, retrieving objects that may have fallen out of the hand during in-hand manipulation. We observe that the reset task is composed of essentially the same reaching, power grasping, and lifting up skills across different objects. Unlike complex manipulation tasks in the in-hand phase, a human operator can provide demonstrations for these skills more conveniently and effectively, overcoming the wide initial state distribution issue due to the fact that objects can fall to anywhere in the environment. As shown in prior work [27], exploration is especially challenging for RL in such settings. Thus, we use behavioral cloning (BC) to train policies for the reset phase from simple demonstrations provided with a 3D mouse and a discrete finger closing command. Note that no demonstrations are used for the actual in-hand reorientation skill (which is difficult to teleoperate), only for the comparatively simpler reset skill, which only requires picking up the object.

**Reward learning via goal images with buffer initialization.** Our aim is to enable our system to learn under assumptions that are reasonable outside of the laboratory: the robot should use the sensors and actuators available to it for all parts of the learning process, including using an autonomous reset policy and eschewing ground truth state estimation (e.g., motion capture) in favor of visual observations that are used to train an end-to-end visuomotor policy. However, this requires us to be able to evaluate a reward function for the in-hand RL training process from visual observations as well, which is highly non-trivial. We therefore use an automated method that uses goal *examples* provided by a person (e.g., positioning the object into the desired pose and placing it on the hand) to learn a success classifier, which then provides a reward signal for RL. Thus, for each in-hand manipulation task $\mathcal{T}_i$, we assume a set $\mathcal{G}_i$ consisting of a few goal images depicting the successful completion of the task. Naïvely training a classifier and using it as a reward signal is vulnerable to exploitation, as RL can learn to manipulate the object so as to fool the classifier [44]. We therefore adapt VICE [44] to address this challenge, which trains an adversarial discriminator pre-defined goal images as positives ($y = 1$) and observation samples from the replay buffer as negatives ($y = 0$). However, it is necessary to adapt this method to handle our buffer initialization approach, since VICE is by design an on-policy [44]. We first summarize the VICE algorithm and the regularization techniques we employ to make it practical for vision-based training, and then discuss how we adapt it to handle buffer initialization.

A common issue with adversarial methods such as VICE is instability and mode collapse. We found strong regularization techniques based on mixup [59] and gradient penalty [60] to be essential to stabilize VICE for learning image-based tasks, and these regularizers additionally aid the RL process by causing the classifier to produce a smoother, more shaped reward signal. The VICE classifier predicts $\log p_\theta(g|o_t)$, the log probability that the observation $o_t$ corresponds to the goal $g$, which can then be used as a reward signal for RL training. The VICE classifier parameterized by $\theta$, $D_\theta$, is then optimized by minimizing a regularized discriminator loss:

$$\mathcal{L}(x; \theta) = \lambda \cdot \mathcal{L}_\lambda(x; \theta) + (1 - \lambda) \cdot \mathcal{L}_{1-\lambda}(x; \theta) + \alpha(\|\nabla_x D_\theta(x)\|_2 - 1)^2 \tag{2}$$

where the input $x$ is a batch of evenly mixed user-defined goal images and observations collected during training, $\mathcal{L}_\lambda$ and $\mathcal{L}_{1-\lambda}$ are the Binary Cross Entropy (BCE) loss terms for mixed-up samples and labels, $\alpha = 10$ is the weight for Gradient Penalty loss.

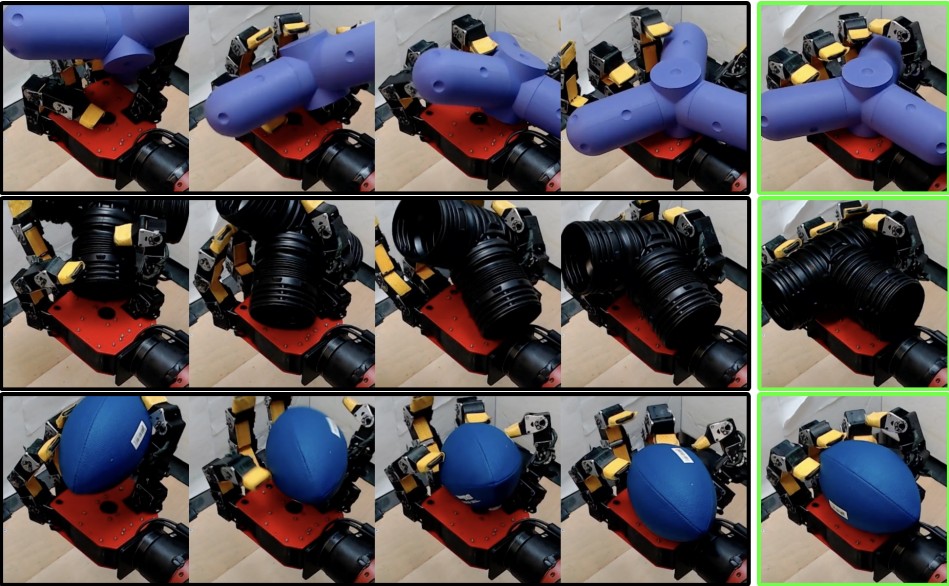

Figure 4: Successful rollouts of in-hand object manipulation policies for the three objects: purple 3-pronged object (Pose B), black T-shaped pipe, and blue football. The boxes on the right (outlined in green) are representative user-provided success state examples for each task. Note that the autonomous pickup policy picks up the object in a variety of different poses across episodes, requiring the in-hand manipulation skill to reorient it into the target pose from many starting configurations.

Applying this method with buffer initialization, where prior data from other tasks and objects is included in the replay buffer, requires additional care. Naïvely, if we train a new VICE classifier with user-provided goal images for the current task as positives, then almost all previous experiences from other tasks and objects are likely to be assigned a negligible reward during training, which would not result in beneficial learning signals for the RL agent. Instead, for tasks from other objects in the prior dataset, rewards are labeled using a task-specific VICE classifier which was trained when that data was collected *for its own task*. These classifier rewards are computed and saved prior to training a new skill, and they remain static throughout training, in contrast to the rewards for online data and offline data from the same object, which depend on the changing VICE classifier.

We hypothesize that initializing the buffer in this way with data from other objects, or other tasks for the same object, will allow the RL algorithm to learn more quickly by transferring knowledge about finger-object interactions, actuation strategies for the hand, and other structural similarities between the tasks. Of course, the degree to which such transfer can happen depends on the degree of task similarity, but as we will show in the next section, we empirically observe improvement from prior data even when transferring to an entirely new object.

## 5 Experimental Results

In our experiments, we aim to study the following questions:

1. Can our system learn dexterous manipulation skills autonomously in the real world?
2. Can prior data from one object improve the sample efficiency of learning new skills with the same object?
3. Can data from different objects be transferred to enable faster acquisition of skills with new objects?

We perform experiments with 3 objects of various shapes and colors: a purple 3-pronged object, a black T-shaped pipe, and a blue football. For each manipulation task, we collected a set of 400 success example images, as described in the Appendix E.

We also provide demonstrations per object for the reset policy to enable in-hand training. We present details of demonstration collection, training procedure, and success rates in Appendix F. Each demonstration takes roughly 30 seconds to collect, totaling less than 2 hours to collect the necessary demonstrations. Please check our website https://sites.google.com/view/reboot-dexterous for videos and further experimental details.

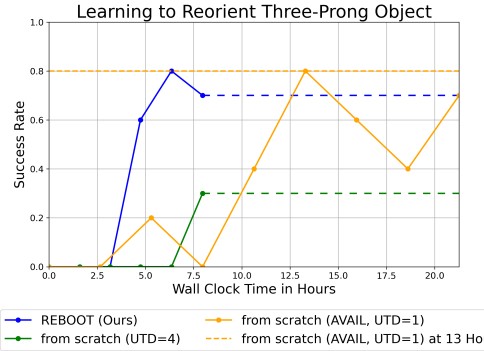

Figure 5: Learning curve showing the performance as a function of training time of reorienting the 3-prong object into different poses. Even though both our method and training from scratch eventually reach a success rate of 80%, our method gets there about two times faster.

Figure 6: Bar plot displaying the training time required for each object to reach their respective target performance. Buffer initialization leads to more than a **2x** speedup across all of the objects compared to training from scratch.

**Task transfer.** To answer Question 1, we evaluated our method on each of the 3 objects with varying amounts of prior data. We first trained a 3-prong object manipulation policy (for a goal pose we call Pose A, shown in Figure 5) without prior data in order to gather data to initialize training for subsequent objects/tasks. We then trained another 3-prong manipulation policy for a different goal pose (Pose B, shown in Figure 4) as well as a T-pipe manipulation policy, both using prior data from the first 3-prong experiment. Finally, we trained a football manipulation policy using the 3-prong and T-pipe experiments as prior data. Our method's success rate is shown in Figure 6, and film strips of various manipulation policy successes during training are shown in Figure 4. Our behavior-cloned reset policy was sufficient as a reset mechanism for in-hand training. Furthermore, our in-hand policies are able to successfully pose the 3-prong and T-pipe objects more than 50% of the time.

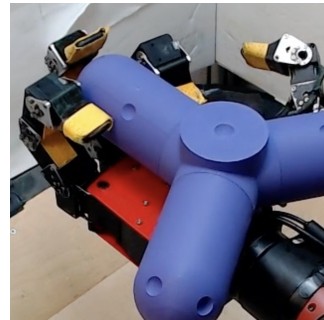

Figure 7: Pose A for the 3-pronged object is approximately $60°$ offset from Pose B, with any leg pointing straight to the wall.

To answer Question 2, we consider the Pose B 3-prong experiment described previously. Since reorienting to both Pose A and B uses the same 3-prong object, we expect the task difficulty to be similar for both poses. A comparison between training Pose A from scratch and training Pose B with a pre-loaded replay buffer is shown in Figure 5. The Pose B experiment with our method outperforms the Pose A experiments training from scratch in terms of training time. Our method reaches 80% success in around 6 hours while training from scratch yields poor performance at that point. It takes more than 10 hours for learning from scratch to achieve a comparable success rate. This suggests that our method can significantly reduce training time when using prior data from the same object for a new manipulation task.

**Object transfer.** To answer Question 3, we consider the T-pipe and football experiments described above. We compare our method to learning from scratch without prior data and display the results in Figure 8. Our method with prior data from other objects is significantly faster than learning from scratch for both objects. For the T-pipe experiments, our method achieves a 60% success rate at 6 hours compared to 13 hours for training from scratch. Furthermore, the from-scratch runs have

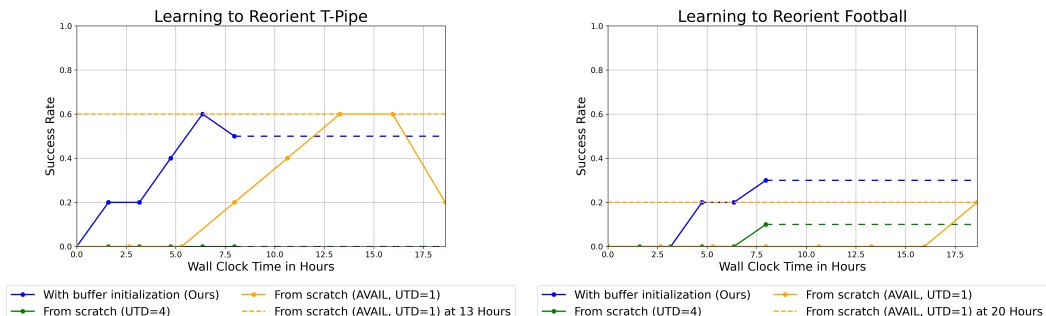

Figure 8: Learning curve showing the performance as a function of training time for the T-pipe and football objects. In both cases, buffer initialization is about two times faster than learning from scratch, though particularly the football object is harder to reorient for all methods.

absolutely no success in evaluation prior to 5 hours of training, while our method achieves some initial success as early as 1 hour into training. The football task appears to be significantly more challenging than the 3-prong and T-pipe tasks, as shown in Figure 8, with no methods performing above a 30% success rate. However, our method still outperforms learning from scratch, achieving a 30% success rate with 5 hours of training; the from-scratch runs required at least 16 hours of training to achieve a lower 20% success rate.

**Ablation Studies.** Finally, we conduct ablation experiments in both simulation and the real world to compare the effects of varying the initial buffer size, the order in which the buffer is initialized, transfer learning from a trained policy, and training for an extended period of time. Results and in-depth analysis are provided in Appendix C and Appendix D.

# 6    Discussion, Limitations, and Future Work

We presented a system for learning in-hand manipulation skills directly by training in the real world with RL, without simulation, and using only onboard sensing from encoders and cameras. Our system enables sample-efficient and autonomous learning by initializing the replay buffer of an efficient online RL method with data from other tasks and even other objects. We extend adversarially-learned classifier-based rewards into this setting to make it possible for users to define tasks with a collection of goal images, and implement automated resets using an imitation-learned reset policy, providing a pipeline for fully autonomous training. The complete system avoids any strong instrumentation assumptions, using the robot's own sensors and actuators for every part of training, providing a proof-of-concept for an efficient real-world RL system that could operate outside of laboratory conditions.

**Limitations:** Our experimental evaluation does have a number of limitations. Although we show that reusing data from one or two prior tasks improves learning efficiency, a more practical general-purpose robotic system might use data from tens, hundreds, or even thousands of skills. Evaluating the potential for such methods at scale might require additional technical innovations, as it is unclear if buffer initialization with very large datasets will be as effective. Additionally, our evaluation is limited to in-hand reorientation skills. While such skills exercise the robot's dexterity and physical capabilities, many other behaviors that require forceful interaction with the environment and other manipulation skills could require a different reset process or might require a different method for reward specification (for example to handle occlusions). Exploring these more diverse skills is an exciting direction for future work. The current manipulation setup is training with fairly robust objects where fragility or wear and tear are not major concerns. As we move to more dexterous tasks, a more directed approach may be required to handle fragile objects or perform tasks that require force-sensitive interaction. Studying how to integrate our system with tactile sensing is another exciting avenue to explore.

**Acknowledgments**

This research was partly supported by the Office of Naval Research (N00014-20-1-2383), and ARO W911NF-21-1-0097. We would like to thank Ilya Kostrikov for the initial versions of the simulator and codebase, and everyone at RAIL for their constructive feedback.

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

## Appendix

## A   Evaluation Success Criteria

We evaluated the trained policy success rate at every 12000 steps for the three real-world in-hand manipulation tasks considered in the paper. The success criteria are defined as follows to stay consistent with the goal images collected to learn the VICE classifiers:

| Task | Success Criteria |
|------|------------------|
| 3-Prong Object Pose A&B | $\mathbb{1}\left\{\left|\theta_{any\_leg} - \theta_{goal\_pose}\right| \leq 5°\right\}$ |
| T-Shaped Pipe | $\mathbb{1}\left\{\left|\theta_{vertical\_leg} - \theta_{goal\_pose}\right| \leq 5°\right\}$ |
| Football | $\mathbb{1}\left\{\left|\theta_{long\_axis} - \theta_{goal\_pose}\right| \leq 5°\right\}$ |

1. 3-Pronged Object:
   - Pose A is successful if any leg is pointing straight forward (to the wall) with less than or equal to 5 degrees deviations.
   - Pose B is successful if any leg is pointing straight backward (to the robot) with less than or equal to 5 degrees deviations.

2. T-Pipe: The T-Pipe is successful if the vertical pipe is pointing straight backward (to the robot) with less than or equal to 5 degrees deviations.

3. Toy Football: The toy football is successful if its long axis is pointing straight to both the wall and the robot with less than or equal to 5 degrees deviations.

For the reset policies, the success criteria are intuitively defined as whether the hand grasps and picks up the object in a ready-to-manipulate pose, such that in-hand training can begin without objects falling out of the palm immediately.

# B   Algorithm Details

In this section, we describe details related to our RL learning algorithms and our imitation learning algorithm and also provide hyperparameters used in experiments for each method.

**REuse Data For BOOTstrapping Efficient Real-World Dexterous Manipulation**

---

**Algorithm 1** REBOOT

---

1: Given: A replay buffer $\mathcal{D}$ with prior data, a set of reset demos $\mathcal{D}_{reset}$, a set of goal images $\mathcal{G}$, and a start state $s_0$.
2: Initialize an empty replay buffer $\mathcal{B}$, RLPD(SAC)[10] with policy $\pi_\psi$ and value function $Q_\psi$, a reset policy $\pi_\phi$, and VICE classifier [44] $D_\theta$
3: Train reset policy $\pi_\phi$ using $\mathcal{D}_{reset}$ via Behavior Cloning
4: **for** iteration $j = 1, 2, ..., T$ **do**
5:     Execute $\pi_\phi$ to perform reset
6:     Execute $\pi_\psi$ in environment, storing data in the online replay buffer $\mathcal{B}$
7:     Update the RLPD's policy and value functions $\pi_\psi, Q_\psi$ using a 50/50 batch of samples from $\mathcal{B}$ and $\mathcal{D}$, assigning reward based on $D_\theta$ using SAC [57].
8:     Update the VICE classifier $D_\theta$ using samples from $\mathcal{B}$ and goal images from $\mathcal{G}$, using eq2.
9: **end for**

---

| Shared RL Hyperparameters | Value |
| --- | --- |
| Shared Images Encoder for DrQ | MobileNetV3-Small-100 with ImageNet-1K weights |
|  | Learned Spatial Embedding |
| Actor Architecture | FC(256, 256) |
|  | FC(256, 19) |
| Critic Architecture | REDQ with 10 Ensembles |
|  | FC(256, 256) |
|  | FC(256, 1) |
| Optimizer | Adam |
| Learning rate | {3e-4} |
| Discount $\gamma$ | 0.99 |
| REBOOT UTD | 4 |
| AVAIL UTD | 1 |
| Target Update Frequency | 1 |
| Actor Update Frequency | 1 |
| Batch size | 256 |
| VICE batch size | 512 (256 Goals + 256 Replay Samples) |

| VICE Classifier Hyperparameters | Value |
| --- | --- |
| Optimizer | Adam |
| Learning rate | {3e-4} |
| Classifier steps per iteration | 1 |
| Mixup Augmentation $\alpha$ | 1 |
| Label Smoothing $\alpha$ | 0.2 |
| Gradient Penalty Weight $\lambda$ | 10 |
| VICE update interval | per episode |
| Classifier Architecture | MobileNetV3-Small-100 with ImageNet-1K weights |
|  | Learned Spatial Embedding |
|  | Dropout(0.5) |
|  | FC(256, 256) $\rightarrow$ LeakyReLU() $\rightarrow$ Dropout(0.1) |
|  | FC(1) |

# C   Ablation Studies

**Order of tasks to initialize the buffer.**    To investigate whether the ordering of tasks to initialize the replay buffer impacts the learning performance in our method, i.e. bootstrapping on one object's or task's data leads to better performance than others, we designed and ran two additional experiments.

The experimental setup follows our method in Figure 5a, where the robot autonomously learns to reorient the 3-prong object to pose B. In the paper, we experimented and reported the performance using replay experiences from pose A's training to bootstrap pose B. Then we bootstrapped the learning of the T-pipe and football using the 3-prong object's data.

We now consider bootstrapping the 3-prong object's pose B learning with replay data from the T-pipe and football training, while keeping the amount of prior data, replay sampling ratio, and UTD the same. We report the evaluation success rate every 12000 steps.

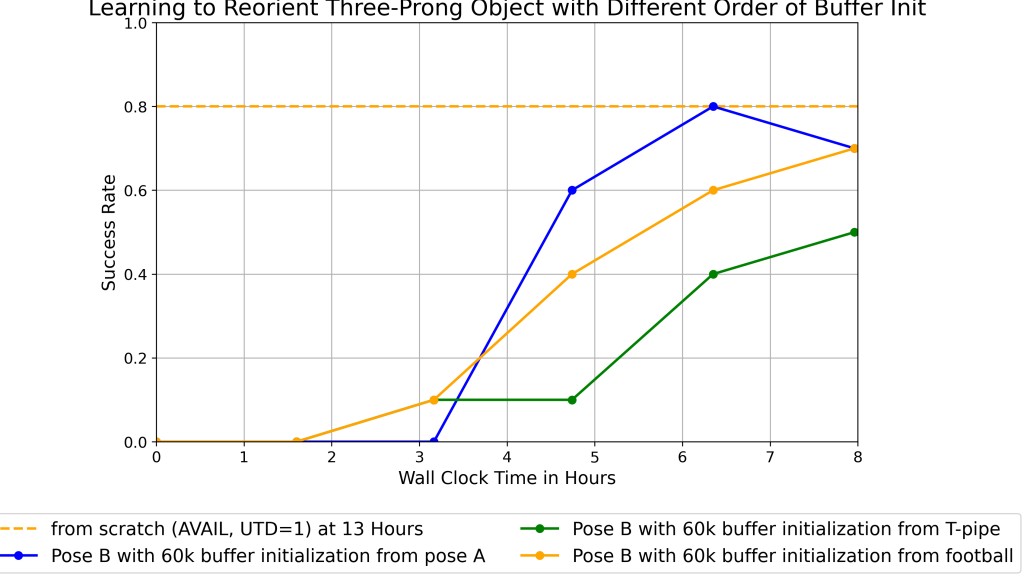

Figure 9: Ablation studying the effect of initializing the replay buffer with prior experience from different objects. Initializing with experience from the same object results in the best performance, but initializing using football experience provides a similar benefit.

For the same task (reorient 3-prong object) under the same training hours, bootstrapping from the same object but different task data yields the best performance, initialized with football task data achieves similar results, T-pipe data's performance follows, and no buffer initialization performed the worst. We note two potentially significant differences between these 3 objects:

1. The T-pipe is fully black colored while the 3-prong object and football are more vividly colored.

2. The in-hand dexterous motions required to solve the tasks are similar between the 3-prong object and the football (planar rotation) but different from the T-pipe (vertical flipping). This can be visualized better on the project website link https://sites.google.com/view/reboot-dexterous).

**Initial buffer size.**   To investigate how different initial replay buffer sizes affect the performance, we performed additional real-world experiments for the 3-prong object reorient task. The experimental setup follows our method in Figure 5a, where the robot is tasked with autonomously learning to reorient the 3-prong object to pose B using buffer initialization from reorienting to pose A. We compare initializing the buffer with 60k vs. 30k randomly selected transitions and apply our method.

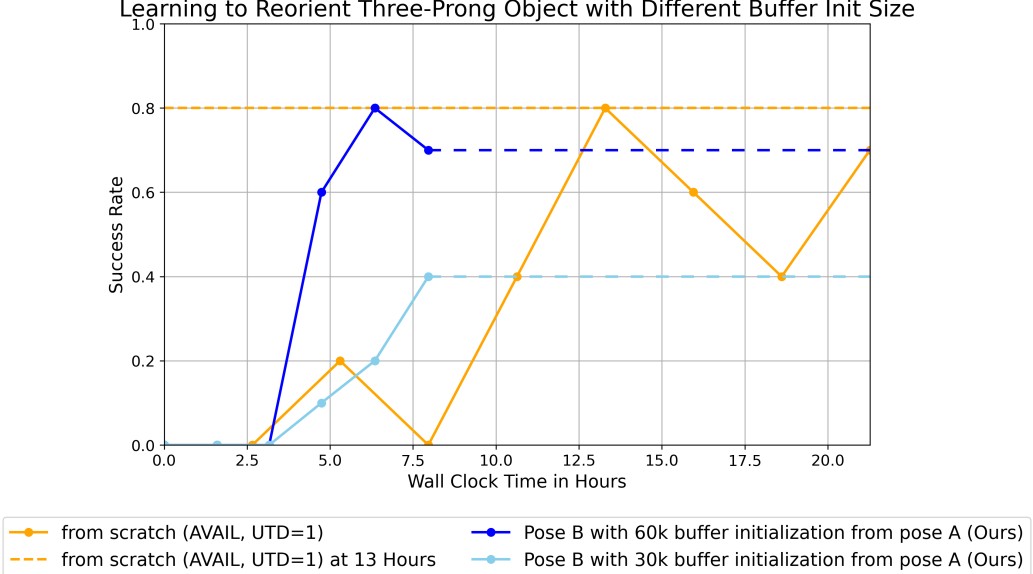

Figure 10: Ablation studying the effect of reducing the amount of data used for buffer initialization (30k vs. 60k transitions pre-loaded into replay buffer). Our result demonstrates that there is some benefit to pre-loading with less data, but the 60k setting still learns considerably faster.

The 30k and 60k transitions used to initialize the replay buffers for these experiments were both sampled uniformly from the same Pose A replay buffer (168k transitions), and both experiments use a 50/50 sampling ratio between prior and new data. However, the run initialized with 60k transitions contains more diverse replay experiences, accelerating the online sample efficiency while achieving a higher success rate under the same training time.

**Comparison to transfer learning.** To compare whether our method is more effective at learning real-world dexterous tasks than alternative approaches such as transfer learning, we ran an additional experiment with the 3-prong object task (Pose B) by transferring a baseline policy (Pose A, UTD=1, no initialization) that is trained for a longer period of time (21 hours, 70% evaluation success rate). We initialized the training by reloading the actor and critics network parameters with the trained checkpoints from the baseline policy and finetuned with the same experimental setups and no replay buffer initialization. We report the evaluation success rate here.

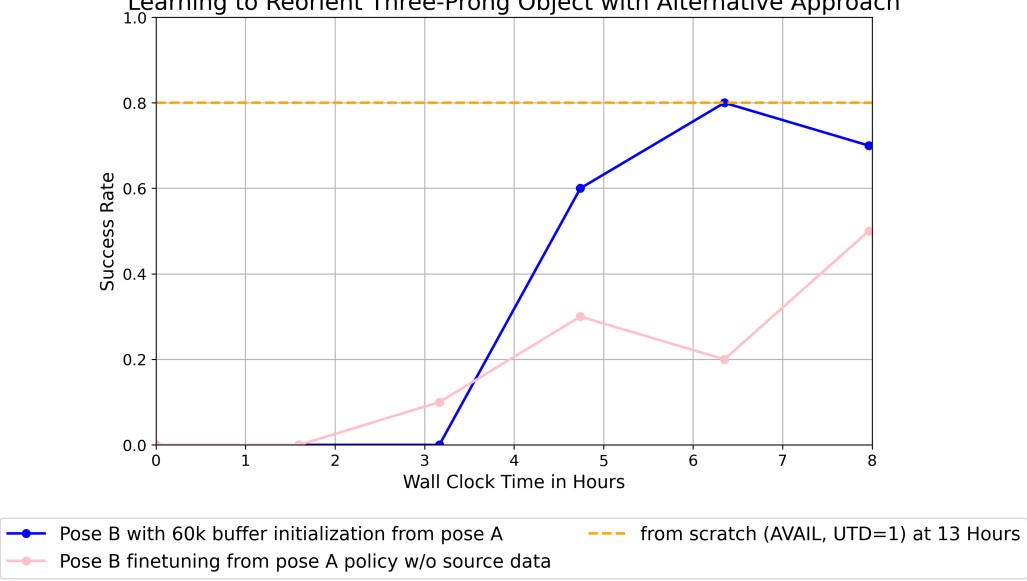

Figure 11: Ablation studying the effect of fine-tuning a previously trained policy for a different goal pose with the same object, rather than pre-loading the replay buffer. We find that pre-loading the replay buffer improves sample efficiency significantly more than fine-tuning an existing policy.

While the policy transfer + finetuning approach outperformed the baseline that learns from scratch under the same training time, our method with buffer initialization still achieves the highest success rate.

**Comparison of all ablations.** Here we visualize a summary of all ablations and comparisons in one plot. Our method is the most sample-efficient among all experiments on reorienting the 3-prong object.

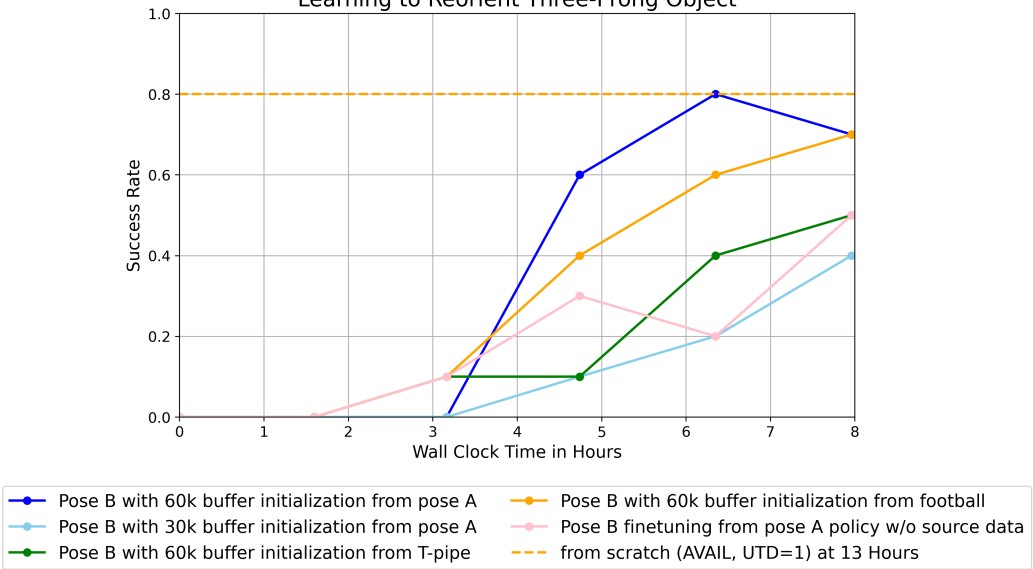

Figure 12: Evaluation plots showing the performance of checkpoints at different points in training for a number of ablation experiments, all learning to reorient the 3-pronged object into Pose B. This figure compares the initial replay buffer size, data used to initialize the replay buffer, and policy initialization ablations against our method. Our method, initialized with 60k transitions from a previous experiment with the same object, clearly learns faster than the ablations.

# D   Longer Training in Simulation

**Simulation Environment:** For testing and iterating our algorithms, we developed a simulation replica of our real robot setup using Mujoco and dm-control. This simulation model consists of the same 16 DoF 4-fingered DHand attached to a 6 DoF Sawyer robot arm as the one built in the real world.

The simulation task considered here is to reposition the 3-pronged object from anywhere on the tabletop back to the center. In this environment, the robot correctly solving the task corresponds to a ground-truth episode reward of -20.

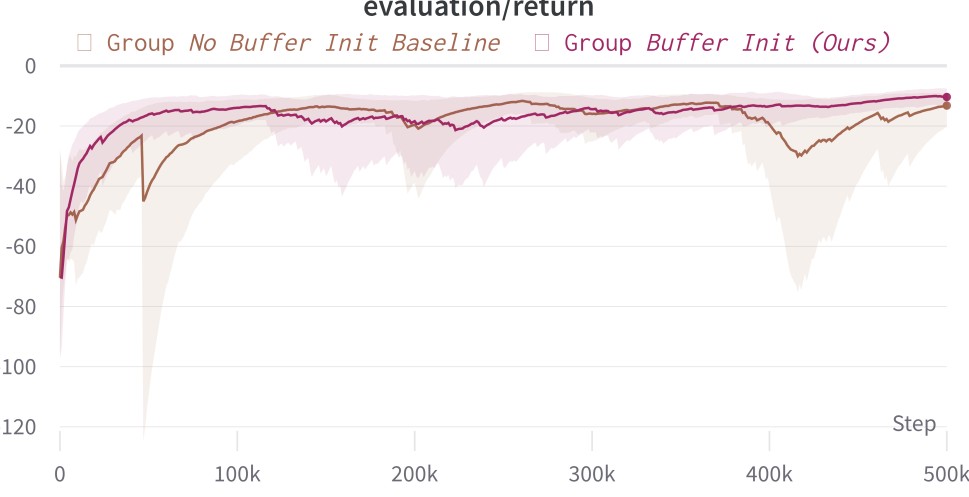

Figure 13: Ground-truth reward vs. training steps for our method and a baseline without buffer initialization in simulation, demonstrating that the performance of our method remains stable after training for a long period of time.

**Results:** Results for simulated experiments are shown in Figure 13. The red line represents the average eval performance of our method across 4 seeds using buffer initialization (UTD=4, 60k transitions initialization, same as real-world), while the brown line represents the average eval performance of the baseline method (UTD=1) w/o buffer initialization across 4 seeds. Both lines are smoothed using an EMA of 0.9. Our method is notably **more sample efficient** at solving the task than the baseline method and is **more stable** at convergence than the baseline when trained up to 500k steps.

# E   Goal Images Collection Procedure

For each task considered in the experiment section, we collect a set of 400 goal images by placing the object in the palm of the robot hand in the desired pose, closing the fingers for 1 second, and executing random actions for 1.5 seconds. We repeat this procedure multiple times, collecting 25 goal images per iteration until we reach 400 total images

# F   Behavior-Cloned Reset Policy Details

We began demonstration collection with the 3-prong object for which we collected 160 demonstrations for the reset policy. We provided only 30 additional demonstrations per new object, for a total of 220 reset demonstrations across all objects. This was sufficient to train a universal reset policy for all objects with a high enough success rate to enable in-hand training.

In most cases, our behavior cloned reset policy is capable of resetting the environment, or at least of making contact with the object, but there are a few states where the policy is unable to pick up or perturb the object in any way. In order to avoid getting stuck attempting unsuccessful resets in these states, we train two different reset policies. One is trained with reset demonstrations for multiple objects, while the other is trained with demonstrations for only the current experiment's object. For example, when running an experiment with the football, one policy is trained using reset demonstrations for the 3-pronged object, the T-shaped pipe, and the football, while the other is trained only with demonstrations for the football. At the start of each training episode, we select the multi-object reset policy with an 80% probability and the single-object reset policy with a 20% probability. Since the policies behave differently due to being trained on different data, states in which one policy might get stuck are unlikely to cause the same issue for the other policy, which enables training to continue even if one of the two policies is sub-optimal.

Here we report the success rate of each reset policy measured when performing evaluations for the in-hand policies.

| Objects | 3-Pronged Object | T-Pipe | Football |
|---|---|---|---|
| Success Rate | 0.608 | 0.667 | 0.367 |

Table 1: Success Rate of Reset Policies

The poor success rate of the toy football could be attributed to its reset success rate. While the 3-pronged object and the T-pipe are more challenging to reorient in-hand due to more complex geometries and more contacts during manipulation than the toy football, it is harder to pick up the toy football by the robot hand due to its slim and small shape. With nearly half the success rate compared to the other two objects, the football in-hand training had fewer opportunities to practice meaningfully. Hence, the football experiment with our method is able to achieve a 30% success rate compared to the near zero success in runs without prior data initialization under the same amount of training time.

