# OpenReview forum: "REBOOT: Reuse Data for Bootstrapping Efficient Real-World Dexterous Manipulation"
_robot-learning.org/CoRL/2023/Conference — CoRL 2023 Poster_

### Official Review · Reviewer_ufd7 · 2023-07-10

**Confidence:** 4
**Originality:** Fair
**Technical Quality:** Good
**Clarity Of Presentation:** Very Good
**Impact:** 3

**Recommendation:**

Weak Accept: I recommend accepting the paper, but will not argue for my recommendation if the majority of other reviewers have a different opinion.

**Review:**

Strengths:
* This paper introduces a new system that learns in-hand reorientation skills autonomously with RL.
* Extensive, convincing hardware experiments are shown.
* The paper is well-structured and written.

Weaknesses:
* In the experimental results section, the emphasis is mainly on showcasing the improvement in sample efficiency resulting from the replay buffer initialization with prior data, which is already known in prior works. It would be interesting to see if a learned policy can handle unseen objects or reorient different objects.
* No state-of-the-art comparison.

**Quality Of The Limitations Section:**

Limitations are addressed clearly

**Questions For Rebuttal:**

In this paper, my main concerns revolve around the weaknesses identified. Furthermore, there are several important pieces of information that require clarification.
* It is essential to explain how the success rate is measured in detail within the experimental results section.
* Regarding Figure 5, it would be helpful to explain the meaning of UTD in the caption.
* Regarding Figure 5(a), based on my understanding, the yellow dotted line represents pose A from scratch (solid yellow line) at 13 wall clock time in hours. However, the current plot is missing vital information as it omits the plots between hours 8 and 13, where the line increases from 0.0 to 0.8. Therefore, I suggest including this information in Figure 5(a) to provide a better understanding of the data

Typos:
- Line 113 - “””
- Line 206 - “additional” to “additionally”

**Robotics Focus:**

Sufficient demonstration on hardware

**Summary Of Paper:**

This paper introduces an approach to enhance the efficiency of reinforcement learning (RL) in tackling complex in-hand reorientation tasks. It addresses various challenges associated with deploying RL methods in real-world scenarios, such as sample efficiency, environment resets, and reward signals. To address these issues, the approach reuses prior data to initialize the replay buffer to improve sample efficiency. It also leverages a behavior cloning (BC) policy to reset the environment for continuous real-world training. Additionally, an adversarial discriminator is used to obtain rewards for online RL training in the real world. The authors conduct real-world experiments on in-hand manipulation of a 3-prong object, a T-shaped pipe, and a football to demonstrate the effectiveness of the approach.

**Summary Of Recommendation:**

After careful consideration of the points raised, I suggest a weak reject. The main concern is the focus of the experiments which primarily demonstrate the benefits of using replay buffer initialization with prior data in terms of sample efficiency, a known insight from prior works. Moreover, the paper lacks insights on how to enhance in-hand reorientation performance. Additionally, it is essential to include a comparison with state-of-the-art approaches to evaluate the proposed method's advancements. Without such a comparison, it is difficult to determine the method's competitiveness and potential impact in the field.

---

### Official Review · Reviewer_puFB · 2023-07-18

**Confidence:** 3
**Originality:** Fair
**Technical Quality:** Good
**Clarity Of Presentation:** Very Good
**Impact:** 3

**Recommendation:**

Weak Accept: I recommend accepting the paper, but will not argue for my recommendation if the majority of other reviewers have a different opinion.

**Review:**

Strengths:

- The paper is well written and easy to understand.
- The quantitative results showcase the sample efficiency of the method when transferred to new goals and objects.
- The end-to-end, reset free training paradigm is exciting to see, and promising for learning real-world robot dexterous manipulation at scale.
- The method requires minimal human intervention and engineering, and does not require hand-engineered rewards.

Weaknesses:

- It is not clear what is the paper’s most significant contribution. It appears that minor variations of multiple pre-established methods have been compiled into a single end-to-end method for learning dexterous manipulation, which is impressive, but for such a systems paper it would be more ideal if there were stronger empirical results showcasing the strength of the proposed system.
- One of the paper’s major focuses is the relative sample efficiency compared to learning from scratch; however, this has already been shown in the prior work cited, and while the proposed method utilizes a novel implementation (images as input applied to robot manipulation domain), that alone does not seem to warrant a strong technical contribution.
- A clear limitation is that the proposed method still requires at least 1 task to be learned from scratch (i.e.: the task that will then seed the resulting data buffer). It is unclear how much this limits the scope of the method – given that even for relatively simple geometries (e.g.: the football), both learning from scratch and even applying REBOOT results in relatively poor success, it is unclear whether the method can be expected to perform well across a wide range of object geometries (especially given that the other two experimental objects share very similar geometries).

**Quality Of The Limitations Section:**

Additional details required

**Questions For Rebuttal:**

Questions:
- Does the ordering of the tasks matter for sample efficiency? E.g.: if the ordering was reversed, where the football was trained first, then the pronged items, would the resulting bootstrapped data be as useful? I could imagine that data from the pronged items may be more useful for downstream tasks as it contains clear features that can be learned.
- How does varying the split between prior / current data for training impact the resulting policy? Is the method sensitive to the relative proportion of prior data used?
- How is success measured? Naively I would imagine the VICE output is used as a surrogate for task success – but if this is the case, how has VICE been validated to produce correct reward annotations for the rollout frames, which may or may not be out of distribution of the frames seen from the previous replay buffer?
- What is the success rate of the learned reset policy? The policy needs to be very robust, and achieve virtually 100% success rate in order to remove the need for a standby human operator. Were there any issues with learning the reset policy?

Technical Concerns:
- Typo in Fig5a: Should dotted y yellow line be Pose B?

**Robotics Focus:**

Sufficient demonstration on hardware

**Summary Of Paper:**

The paper proposes a novel system for learning real-world dexterous in-hand manipulation policies in an end-to-end fashion (REBOOT). By leveraging a combination of buffer initialization and reset-free learning techniques, the proposed method is able to achieve desired target goals with greater sample efficiency compared to learning policies from scratch. The method is evaluated against three different objects and is shown to have success when transferred across the different objects.

**Summary Of Recommendation:**

The proposed method clearly exhibits greater sample efficiency compared to learning individual in-hand policies from scratch by leveraging prior task data. Moreover, the method requires minimal assumptions, and does not require manual resets or rewards. However, the overall novelty of the system remains unclear, with more analysis and discussion needed to better justify the technical contributions of the overall system.

---

> ### Author Response · Authors · 2023-08-12
> **Friendly Reminder**
>
> Dear Reviewer,
>
> We wish to remind you that we have submitted our rebuttal in response to the feedback and suggestions you provided. It would be highly valuable for us if you could review it. We have specifically addressed the critical points you highlighted, such as ablation experiments and evaluation criteria. Please let us know whether we have adequately addressed your concerns.
>
> We genuinely appreciate the time and effort you've put into reviewing our work. If you find our responses satisfactorily address your concerns and feedback, we kindly invite you to consider adjusting the score accordingly.
>
> Best regards,
> The Authors

---

> ### Author Response · Authors · 2023-08-14
> **Deadline Approaching Reminder**
>
> Dear Reviewer,
>
> We wish to remind you that we have submitted our rebuttal in response to the feedback and suggestions you provided. It would be highly valuable for us if you could review it. We have specifically addressed the critical points you highlighted, such as ablation experiments and evaluation criteria. Please let us know whether we have adequately addressed your concerns.
>
> We genuinely appreciate the time and effort you've put into reviewing our work. If you find our responses satisfactorily address your concerns and feedback, we kindly invite you to consider adjusting the score accordingly.
>
> Best regards,
> The Authors

---

### Official Review · Reviewer_7J5e · 2023-07-19

**Confidence:** 4
**Originality:** Good
**Technical Quality:** Fair
**Clarity Of Presentation:** Good
**Impact:** 3

**Recommendation:**

Weak Accept: I recommend accepting the paper, but will not argue for my recommendation if the majority of other reviewers have a different opinion.

**Review:**

This work presents an interesting and practical exploration, confirming that leveraging prior non-task specific data can significantly enhance skill-learning efficiency in real-world dexterous manipulation tasks.

One notable limitation is its confinement to image-based domains, where different tasks share the same observation space (dimension). This commonality enables the utilization of prior data effectively, possibly due to the properties of image signals. Additionally, the existence of shared visual features in the same dexterous manipulator allows for the transfer of visual-based control skills across various tasks.

However, to further enhance the work's credibility, a more comprehensive analysis could be beneficial. Additional details regarding this aspect can be found in the questions section. Addressing these points would strengthen the overall contribution of the study.

For details please refer to the question part.

**Quality Of The Limitations Section:**

Additional details required

**Questions For Rebuttal:**

My main concern is that the experiment part are not complete.

Q1-1: Why 50/50 of current task buffer and prior tasks buffer? Maybe practically it works the best, but is there any comparison experiment that supports this choice?

Q1-2: Is this 50/50 ratio fixed during the training? if so, could there be any adaptive strategy for mixing the buffer? e.g., gradually reduce the usage of previous (other tasks') buffer.

Q2: I think several ablation study is missing, e.g.,: How does initialization buffer size affect the performance? In another word, for example, how 10/50/100 episode-size initial buffer affect the training performances?

Q3: As this approach uses prior tasks' skills to initialize the buffer, I would like to see the comparison with transfer learning and meta-learning based solutions.

**Robotics Focus:**

Sufficient demonstration on hardware

**Summary Of Paper:**

This practical study explores the role of replay buffer initialization in accelerating RL skill learning, particularly by leveraging prior data from other tasks.

More specifically, this work also proposed a modified visual reward learning algorithm considering the random replay buffer initialization, which is to keep the VICE reward from other tasks static.

Real-world experiments validate the effectiveness of this simple yet powerful technique, demonstrating significant improvements in sample efficiency when learning new skills, even across different object domains.

**Summary Of Recommendation:**

I really appreaciate the idea of using prior non-tasl related data to accelerate the skill learning. It also surprisingly accelerate the skill learning practically. However, more ablation studies are need to make this work complete.

-----------------------------------------

The authors added the additional ablation studies and comparison experiments. After the Rebuttal I would change my rating to accept

---

> ### Author Response · Authors · 2023-08-12
> **Friendly Reminder**
>
> Dear Reviewer,
>
> We wish to remind you that we have submitted our rebuttal in response to the feedback and suggestions you provided. It would be highly valuable for us if you could review it. We have specifically addressed the critical points you highlighted, such as ablation experiments and other approaches. Please let us know whether we have adequately addressed your concerns.
>
> We genuinely appreciate the time and effort you've put into reviewing our work. If you find our responses satisfactorily address your concerns and feedback, we kindly invite you to consider adjusting the score accordingly.
>
> Best regards,
> The Authors

---

> ### Author Response · Authors · 2023-08-14
> **Deadline Approaching**
>
> Dear Reviewer,
>
> We wish to remind you that we have submitted our rebuttal in response to the feedback and suggestions you provided. It would be highly valuable for us if you could review it. We have specifically addressed the critical points you highlighted, such as ablation experiments and evaluation criteria. Please let us know whether we have adequately addressed your concerns.
>
> We genuinely appreciate the time and effort you've put into reviewing our work. If you find our responses satisfactorily address your concerns and feedback, we kindly invite you to consider adjusting the score accordingly.
>
> Best regards,
>
> The Authors

---

### Official Review · Reviewer_tREV · 2023-08-04

**Confidence:** 4
**Originality:** Fair
**Technical Quality:** Fair
**Clarity Of Presentation:** Good
**Impact:** 3

**Recommendation:**

Weak Accept: I recommend accepting the paper, but will not argue for my recommendation if the majority of other reviewers have a different opinion.

**Review:**

Overall, the real world demonstrations of this robot platform and approach are one of the strongest points of this method. They seem to show fairly strong results in the four tasks they propose, learning within a few hours. However, for a method like this, training in simulation for speed of testing and verifying that their method can scale to the chosen tasks is fairly critical. This is especially true since some of the experimental runs require hours of real-time training to evaluate a single model, which is bound to be bottle-necked by robot hours of operation. It is highly possible that the unusual drops in performance as more training time progresses In Figures 5 and 6 is an artifact of the object resetting approach used, but this goes unmentioned in the experimental section.
Additionally, the link does not seem to work.

**Quality Of The Limitations Section:**

Additional details required

**Questions For Rebuttal:**

The experiments aim to answer whether this system can learn skills autonomously, whether using REBOOT improves the sample efficiency of learning new skills, and whether faster acquisition of skills is possible. While these are indeed essential questions, another one worth investigating is whether, given enough training time, if learning from scratch end-to-end (with the reset policy), can perform better than when regularized with tasks which may limit upperbound performance. This would be a good additional question to validate, and can be done with the existing runs, with longer training runs on the robot (or in simulation).

**Robotics Focus:**

Sufficient demonstration on hardware

**Summary Of Paper:**

The paper, titled REBOOT, proposes a method for reusing data for bootstrapping real-world dexterous manipulation, and creates an autonomous pipeline to reset and collect data that can be used across multiple tasks for training task-specific policies. This includes skills such as in-hand manipulation of a triangular valve, a T-shaped pipe, and a toy football. Their method trains a reset policy (via imitation learning) and an RL policy in tandem to train a system that can autonomously collect data in the real world.

**Summary Of Recommendation:**

After the rebuttal period, the authors produced more results indicating that their approach is not handicapped by the amount of training they performed on the real robot. After adding simulated experiments, they confirmed that their method's results were not limited by the robot hours, and that REBOOT has a stronger effect on the learned policy success rate than policy transfer and fine-tuning. For this reason, I recommend the paper as a weak accept, as it produces some interesting insights for reset-free methods and fine-tuning for dexterous grasping tasks.

---

> ### Author Response · Authors · 2023-08-05
> **Response to Reviewer tREV (Initial Clarification)**
>
> Thank you for the comments and the improvements that you have suggested. While we are conducting more experiments to strengthen the qualities of our comparisons and analysis, we would like to clarify the website link in the paper https://sites.google.com/view/reboot-dexterous/ is working and contains many additional training videos and experimental setups and details. We apologize for the inconvenience that the link in OpenReview abstract is typed with Latex notation, which causes OpenReview to create an incorrect hyperlink.

---

> ### Author Response · Authors · 2023-08-14
> **Simulation Runs Update & Deadline Approaching Reminder**
>
> Dear Reviewer,
>
> We wish to update you on the latest results of our simulation experiments. As described in the rebuttal post, we ran and evaluated the performance of our method (utd=4 with 60k steps buffer init) and the baseline method (AVAIL, utd=1 w/0 buffer init) in a simulation replica of our real robot. With the 3-pronged object tabletop reposition task, our method is not only **more sample efficient** at solving the task than the baseline method and is **more stable at convergence** (episode return greater than or equals to -20) than the baseline when trained up to 500k steps. The plot and comparison are available on the [project website](https://sites.google.com/view/reboot-dexterous/#h.bghi6zk4ksuf) and also in the pdf appendix.
>
> We also wish to remind you that we have submitted our rebuttal in response to the feedback and suggestions you provided. It would be highly valuable for us if you could review it. We have specifically addressed the critical points you highlighted, such as ablation experiments and evaluation criteria. Please let us know whether we have adequately addressed your concerns.
>
> We genuinely appreciate the time and effort you've put into reviewing our work. If you find our responses satisfactorily address your concerns and feedback, we kindly invite you to consider adjusting the score accordingly.
>
> Best regards,
>
> The Authors

---

> ### Author Response · Authors · 2023-08-15
> **Kind Follow-Up on Our Rebuttal Responses**
>
> Dear Reviewer,
>
> We hope this message finds you well. We'd like to express our gratitude to you for the time you spent reviewing our work and rebuttal. In case you missed our previous note, we would truly appreciate it if you could review our responses to the feedback provided. The additional real-world and simulation experiments we added have addressed two other reviewers' concerns positively. We hope that the more solidified version of our work also addresses your concerns.
>
> Our primary goal is to ensure that we've satisfactorily addressed any concerns you may have raised. Your feedback and insights are invaluable to our work, and we sincerely appreciate the time and effort you've dedicated to this review process.
>
> Best regards,
>
> The Authors

---

> ### Comment · Reviewer_tREV · 2023-08-15
> **Updated Review**
>
> Upon reviewing the rebuttal, I appreciate the lengths the authors took to address the core issues with the method, namely that the performance shown in the paper was limited by robot hours, and their results in simulation have contributed to a more complete set of results. Specifically, they show in their additional experiments on increasing initial buffer size that this has a stronger effect on the learned policy success rate than policy transfer and fine-tuning, which justifies the validity of their approach as a method for "reset-free" reinforcement learning methods.

---

> > ### Author Response · Authors · 2023-08-15
> > **Thank you note**
> >
> > Dear Reviewer,
> >
> > Thank you again for the detailed feedback and suggestions! Your input truly helped us improve the paper. We will make sure to include the complete ablation studies and comparisons in the final version of the paper.
> >
> > Best Regards,
> >
> > The Authors

---

### Decision · Program_Chairs · 2023-08-30

**Decision:**

Accept (Poster)

**Comment:**

This paper presents an efficient system for learning dexterous manipulation skills through sample-efficient RL and replay buffer bootstrapping. After discussion and clarification during the rebuttal phase by the authors, all the reviewers have issued 3 weak accepts and 1 accept.